# Traffic Flow Prediction Based on Hybrid Deep Learning Models Considering Missing Data and Multiple Factors

**Wenbao Zeng** [1,2,3], **Ketong Wang** [1,2,3], **Jianghua Zhou** [1,2,3] **and Rongjun Cheng** [1,2,3,*]

1 Faculty of Maritime and Transportation, Ningbo University, Ningbo 315211, China; 18859197609@163.com (W.Z.); 206000066@nbu.edu.cn (K.W.); zhoujianghua@nbu.edu.cn (J.Z.)
2 Jiangsu Province Collaborative Innovation Center for Modern Urban Traffic Technologies, Nanjing 210096, China
3 National Traffic Management Engineering and Technology Research Centre, Ningbo University Sub-Centre, Ningbo 315211, China
* Correspondence: chengrongjun@nbu.edu.cn

**Abstract:** In the case of missing data, traffic forecasting becomes challenging. Many existing studies on traffic flow forecasting with missing data often overlook the relationship between data imputation and external factors. To address this gap, this study proposes two hybrid models that incorporate multiple factors for predicting traffic flow in scenarios involving data loss. Temperature, rainfall intensity and whether it is a weekday will be introduced as multiple factors for data imputation and forecasting. Predictive mean matching (PMM) and K-nearest neighbor (KNN) can find the data that are most similar to the missing values as the interpolation value. In the forecasting module, bidirectional long short-term memory (BiLSTM) network can extract bidirectional time series features, which can improve forecasting accuracy. Therefore, PMM and KNN were combined with BiLSTM as P-BiLSTM and K-BiLSTM to forecast traffic flow, respectively. Experiments were conducted using a traffic flow dataset from the expressway S6 in Poland, considering various missing scenarios and missing rates. The experimental results showed that the proposed models outperform other traditional models in terms of prediction accuracy. Furthermore, the consideration of whether it is a working day further improves the predictive performance of the models.

**Keywords:** traffic flow prediction; missing data; data imputation; KNN; PMM; BiLSTM

## 1. Introduction

Traffic congestion has now become a common problem in many large cities [1]. When traffic congestion occurs, it is accompanied by negative impacts such as economic losses, increased difficulty in managing traffic and air pollution [2]. Traffic congestion can be greatly reduced by accurately predicting traffic flows and helping travelers to make informed route choices. On the other hand, traffic management department can guide traffic with traffic flow forecasts and thus provide a comfortable route for travelers. For the entire road network, relieving traffic congestion requires a sound Intelligent Transport Systems (ITS) of which traffic flow prediction is a key factor [3]. Efficient and accurate traffic flow prediction can provide data to support the working of ITS.

In recent decades, traffic forecasting has been studied involving traffic flow prediction, travel time forecasting, speed forecasting, and so on. In this regard, traffic flow prediction can be divided into short-term and long-term ones. Short-term traffic flow forecasting is more widely studied because of its greater applicability. Vlahogianni et al. [4] has shown that short-term traffic predicting methods fall into two main categories, namely classical statistical methods and computational intelligence (CI) methods. The most commonly used classical statistical methods are the autoregressive integrated moving average (ARIMA) model and its refinements [5,6]. But these methods are usually designed for small data sets and are not suitable for dealing with complex and dynamic time series data. Currently,

the most commonly used forecasting methods are CI methods, such as artificial neural networks (ANNs) [7], convolutional neural networks (CNNs) [8], long short-term memory (LSTM) [9] neural networks and graph convolutional network (GCN) [10].

Different external factors may have an impact on traffic flow. It is common for some scholars to add relevant factors within the model in the hope of improving prediction accuracy. Zhang et al. [11] used a multi-factor gated recurrent unit (GRU) for traffic flow prediction, incorporating factors such as precipitation, average wind speed, maximum temperature, minimum temperature and weather types into the model. The results proved that the multi-factor GRU model provided better prediction results. Chen et al. [12] proposed the attentive attributed recurrent graph neural network (AARGNN) which predicts short-term traffic flow considering both static and dynamic factors. Experiments on real-world datasets showed that the proposed method outperforms all baseline methods. He et al. [13] proposed the multi-graph convolutional-recursive neural network (MGC-RNN). They creatively generated five correlation diagrams with multiple external factors as model inputs to predict subway passenger flow.

Most of the existing studies simply added external factors into prediction models and drew conclusions that these external factors can improve prediction accuracy. But He has shown that not all factors improve the accuracy of predictions [13]. Few studies have focused on the relationship between external factors and prediction accuracy.

The majority of predictive models rely on the completeness of the data set. However, due to some unavoidable factors, the information collected by the sensors may have missing data. To reduce the impact of missing data, data imputation has been used in prediction models. Such prediction models are classified as hybrid and fusion models. Khan et al. [14] used multiple imputation methods and a combination of neural networks to predict the daily average traffic flow and hourly traffic volume. The combination of LSTM and mean-fill models was eventually found to provide the best prediction results. Traffic flows are significant not only temporally but also spatially. Tensors provide a simple and effective approach to represent spatio-temporal traffic flows. Therefore, several scholars have adopted a tensor-based approach to complement traffic flow interpolation and prediction [15–17]. The graph Laplacian method offers an efficient approach for extracting spatio-temporal information, which can be combined with LSTM to achieve accurate predictions in scenarios with missing data [18]. Zhao et al. [19] proposed two mean imputation methods combined with LSTM to achieve traffic flow prediction under three missing modes. All of the above studies are the combination of imputation methods and prediction models to achieve prediction.

On the other hand, fusion models have been proposed which can perform both data imputation and traffic flow prediction. Cui et al. [20] proposed an LSTM structure with imputation units (LSTM-I) to fill in the missing values in the input data. The two-layer bidirectional LSTM-I achieved high accuracy in attribution and prediction under different missing patterns, even with 80% of the data missing. However, LSTM-I is limited to extracting only temporal features and lacks the ability to capture spatial feature. For better consideration of spatial factors, graph convolution is widely used. Cui et al. [21] proposed graph Markov network (GMN) and spectral graph Markov network (SGMN) with spectral graph convolution operations. The GMNs and SGMNs were experimentally shown to perform well in terms of prediction accuracy and efficiency. While GMN and SGMN lack the capability to extract time series features, GRU excels at capturing time-varying features. Combining convolutional operations with GRU neural networks provides a distinctive advantage in spatio-temporal prediction. Zhang and Dong have further enhanced this approach to enable accurate predictions in various data scenarios [22,23].

In these studies, both hybrid and fusion models ignore the impact of multiple factors on traffic flow. There is no doubt that multiple factors can have some influence on traffic flow. Until now, there are relatively fewer studies that consider multiple factors in both imputation and forecasting models. Due to this reason, we incorporate multiple factors

into the imputation and forecasting methods to further improve accuracy and reduce the impact of missing data.

Two hybrid models have been proposed to explore the impact of multiple factors on prediction accuracy under different missing scenarios with different missing rates as follows:

1.  The influence of multiple factors was considered to enhance the interpretability of feature selection;
2.  Two prediction models have been proposed for the missing data scenario, and multivariate data are used to improve accuracy in the missing data scenario;
3.  To be more realistic, a random missing scenario and a non-random missing scenario were set up and the impact of different missing scenarios on prediction accuracy is explored.

The rest of the paper is organized as follows. Section 2 presents the structure of the proposed models. Section 3 describes the data sources and correlation analysis. Section 4 analyzes the prediction results of different combinations of models incorporating different factors. Section 5 highlights the conclusion of this paper and the outlook for future work.

## 2. Methodologies

In this chapter, the imputation models and the prediction models are introduced firstly, and then the proposed models are presented in detail.

### 2.1. Imputation Models

#### 2.1.1. K-Nearest Neighbor

KNN is a basic machine learning algorithm for classification and regression. The central idea is to use the samples that are closest to the unknown samples to carry out the classification and prediction. Changes in traffic flow can be influenced by external factors and the KNN algorithm can interpolate missing traffic data by picking up the traffic flows that are most similar to the external factors. The advantage of this algorithm is that it takes into account the impact of external factors on the traffic flow. The KNN algorithm can be implemented in the following two steps [24].

Step 1: In this step, $k$ samples are selected mainly according to their distances. Too large or too small $k$ will increase the error. Through repeated experiments and verifications, we found that when $k = 5$, the imputation effect is the best. The closer the two samples are, the higher the similarity, and vice versa. The distance is measured via Euclidean distance, and its calculation formula is shown in Equation (1):

$$d = \sqrt{\sum_{j=1}^{m}(x_j - y_j)^2} \tag{1}$$

where $m$ represents the number of influencing factors other than traffic flow, $x_j$ is the value of the $j$-th factor for missing data and $y_j$ is the value of the $j$-th factor of the complete data.

Step 2: After getting all the distances between the missing data and the complete data, select the $k$ complete data closest to the missing data, and calculate the average of the $k$ traffic flows to fill in the missing values.

#### 2.1.2. Predictive Mean Matching

Multiple imputation (MI) as an effective data imputation method was first proposed by Rubin in 1977 [25]. PMM was proposed by Little and has been refined to become one of the most classical and commonly used MI algorithms [26]. PMM is based on a complete data set, regressed on the corresponding variables, followed by a regression model to obtain imputed values which are filled by taking the mean of multiple imputed values. The PMM algorithm proceeds as follows [27]:

Step 1: Let the sample size be $n$. The number of samples with no missing data is $n_{obs}$ and the number of samples with missing data is $n_{mis}$. $Y_{obs}$ and $Y_{mis}$ denote the existing observations and missing values in $Y$, respectively. $X = (X_1, X_2, \ldots X_k)$ is a set of fully observed covariates, which includes $X_{obs}$ and $X_{mis}$, with $X_{mis}$ corresponding to the missing part observed in $Y$.

Step 2: Use $Y_{obs}$ and $X_{obs}$ to calculate the least squares estimates $\hat{\beta} = (\hat{\beta}_0, \hat{\beta}_1, \hat{\beta}_2 \ldots \hat{\beta}_k)$; errors $\varepsilon$, and residual variances $\hat{\sigma}^2$ can be computed as:

$$Y = \beta_0 + \beta_1 X_1 + \beta_2 X_2 + \ldots + \beta_k X_k \tag{2}$$

Step 3: $\hat{\sigma}^2$ is subject to a $\chi^2$ distribution with degree of freedom $n_{obs} - k - 1$. Take a random number $g$ from the $\chi^2$ distribution, and obtain the random observation $\sigma^2$.

$$\sigma^2 = \hat{\sigma}^2 (n_{obs} - k - 1)/g \tag{3}$$

Step 4: Draw $\beta^*$ from a multivariate normal distribution centered at $\hat{\beta}$ with covariance matrix $\sigma^2$.

Step 5: The fitted and predicted values are calculated as follows:

$$Y_{obs} = \hat{\beta}_0 + \hat{\beta}_1 X_1 + \hat{\beta}_2 X_2 + \ldots + \hat{\beta}_k X_k \tag{4}$$

$$Y_{mis} = \beta_0^* + \beta_1^* X_1 + \beta_2^* X_2 + \ldots + \beta_k^* X_k \tag{5}$$

Step 6: Calculate the distance $\Delta_i$ between $\hat{Y}_{obs,i}$ and $\hat{Y}_{mis}$ as follows:

$$\Delta_i = \left| \hat{Y}_{obs,i} - \hat{Y}_{mis} \right| \tag{6}$$

where $i = 1, 2, 3 \ldots n_{obs}$.

Step 7: Select the smallest $\Delta_i$ corresponding to the $\hat{Y}_{obs,i}$ as the imputation value. Repeat step 2–7 times and choose the mean.

### 2.2. Prediction Models

#### 2.2.1. Long Short-Term Memory

To solve the gradient vanishing and gradient exploding problems, Hochreiter and Schmidhuber proposed the LSTM neural network on the basis of recurrent neural networks (RNNs) [28]. As shown in Figure 1a, an LSTM with a unique chain structure is able to capture the regular characteristics of time series and thus achieve time series prediction. As the parameters of each structure are independent, the gradient disappearance and gradient explosion problems are effectively avoided. In Figure 1b, the internal structure of the LSTM is shown in detail and is composed of forget gate $f_t$, input gate $i_t$, output gate $o_t$, memory cell $c_t$ and current output $h_t$. The output value of its previous unit $h_{t-1}$, the cell state of the previous unit $c_{t-1}$ and the input data $x_t$ are used as the input of the current unit. LSTM can be described using the following formulas [29]:

$$f_t = \sigma \left( W_f x_t + U_f h_{t-1} + b_f \right) \tag{7}$$

$$i_t = \sigma (W_i x_t + U_i h_{t-1} + b_i) \tag{8}$$

$$o_t = \sigma (W_o x_t + U_o h_{t-1} + b_o) \tag{9}$$

$$\widetilde{c}_t = \tanh (W_c x_t + U_c h_{t-1} + b_c) \tag{10}$$

$$c_t = f_t \odot c_{t-1} + i_t \odot \widetilde{c}_t \tag{11}$$

$$h_t = o_t \odot \tanh(c_t) \tag{12}$$

where $W_f$, $U_f$, $W_i$, $U_i$, $W_o$, $U_o$, $W_c$ and $U_c$ are weight matrices, $b_f$, $b_i$, $b_o$ and $b_c$ are bias vectors, $\odot$ is the Hadamard product, and $\sigma$ and tanh are activation functions. Their formulas are as follows:

$$\sigma(x) = \frac{1}{1 + e^x} \tag{13}$$

$$\tanh(x) = \frac{e^x - e^{-x}}{e^x + e^{-x}} \tag{14}$$

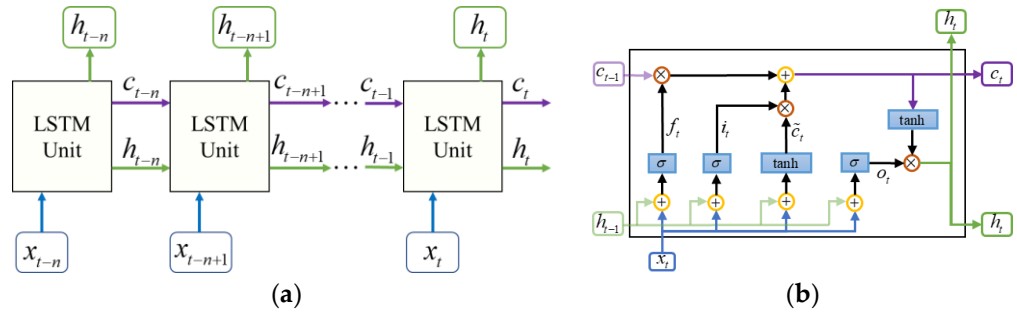

**Figure 1.** The structure of the LSTM. (**a**) The overall structure of the LSTM. (**b**) Internal structure of the LSTM unit.

The final predicted value $\hat{y}_{t+1}$ after passing through the fully connected layer is:

$$\hat{y}_{t+1} = \tanh(h_t W_y + b_y) \tag{15}$$

where $W_y$ is weight matrix, and $b_y$ is bias vector.

### 2.2.2. Bidirectional Long Short-Term Memory

LSTM is a forward training model that can only extract forward time series information, and reverse information is not well extracted. To this end, Graves and Schmidhuber have proposed a BiLSTM that combines reverse LSTM and forward LSTM [30]. Because it extracts time series information in both directions, it has more advantages in terms of prediction. The structure of the BiLSTM is shown in Figure 2. In the figure, $x$ is fed into the forward and reverse LSTM to obtain the output of the LSTM in different directions, which are combined to obtain the final prediction $y$. The final predicted value is calculated as follows:

$$\hat{y}_{t+1} = \tanh\left(\overrightarrow{h}_t W_y + \overleftarrow{h}_t U_y + b_y\right) \tag{16}$$

where $\overrightarrow{h}_t$ and $\overleftarrow{h}_t$ denote the output result of the forward LSTM and the one of the reverse LSTM, respectively. $W_y$ and $U_y$ are weight matrices and $b_y$ is bias vector.

### 2.3. Proposed Hybrid Model

In order to solve the problem of missing data, we combine the imputation module and the prediction module to complete the traffic flow prediction in the presence of missing data. PMM, KNN and BiLSTM are combined to form P-BiLSTM and K-BiLSTM, respectively, as shown in Figure 3 with their model structures. In the models, there are two main modules, namely the imputation module and the prediction module. The imputation module completes the imputation of the data, while the prediction module predicts the traffic flow.

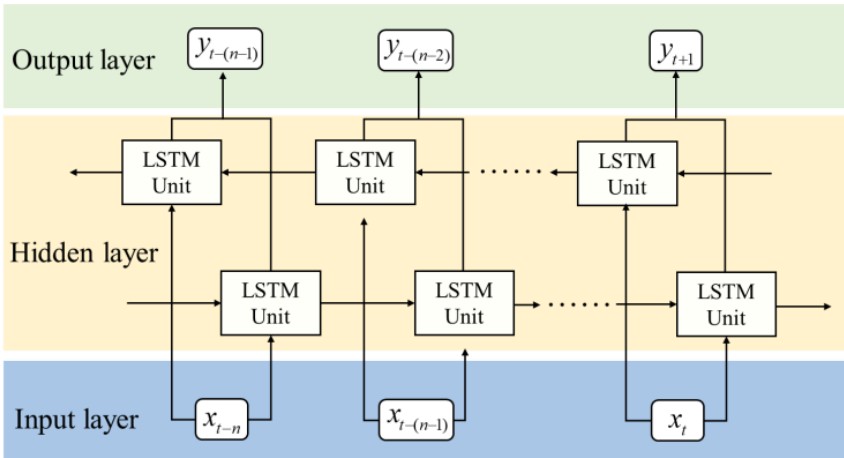

**Figure 2.** The structure of the BiLSTM.

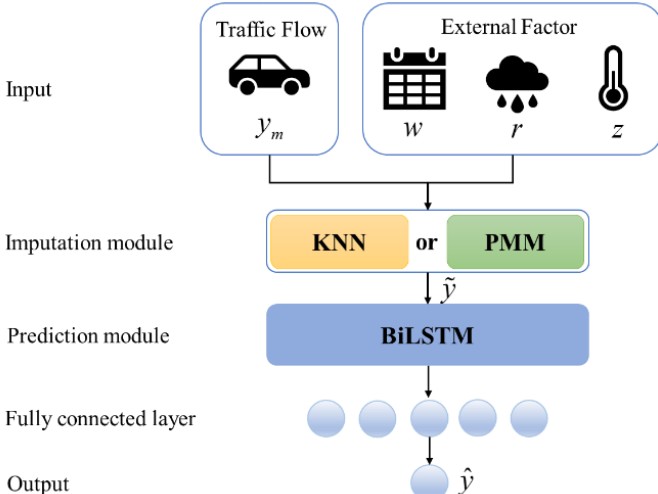

**Figure 3.** The structure of K-BiLSTM and P-BiLSTM models.

The input is divided into two parts: the traffic flow $y^m$ with missing data and three multiple factors, including whether it is a weekday $w$, the rainfall intensity $r$ and the temperature $z$. The expressions for the inputs at time $t$ is as follows:

$$x_t = \begin{pmatrix} y_t^m & w_t & r_t & z_t \\ y_{t-\Delta}^m & w_{t-\Delta} & r_{t-\Delta} & z_{t-\Delta} \\ \vdots & \vdots & \vdots & \vdots \\ y_{t-n\Delta}^m & w_{t-n\Delta} & r_{t-n\Delta} & z_{t-n\Delta} \end{pmatrix} \tag{17}$$

where $\Delta$ denotes the time interval and $t - n\Delta$ denotes t indicates the traffic flow and the statistics of each factor for the previous $n$ time periods.

Traffic flow is normalized to a range of $[0, 1]$. A binary variable is used to indicate whether it is a weekday, with "1" representing workday and "0" indicating the weekend.

The $x_t$ with missing values is fed to the KNN or PMM to obtain the complete traffic flow data $\tilde{y}$ via the imputation module, followed by the flow and external factors together into BiLSTM via the fully connected layer to obtain the final output $\hat{y}$.

## 3. Data Analysis

### 3.1. Data Sources

The traffic data used for predication are from the permanent traffic counting station located on the expressway S6 in the Tricity agglomeration area in Poland. The data covers a three-year period from 2014 to 2017, as well as traffic in one direction (southbound). Tricity Bypass Road (expressway S6) is the eastern end segment of the Polish National Road No. 6 which runs along the Baltic coast between the cities of Szczecin and the Tricity Metropolitan Area, comprising the cities of Gdansk, Sopot and Gdynia. As shown in Figure 4, red pentagram indicates counting stations and blue line represents expressway S6.

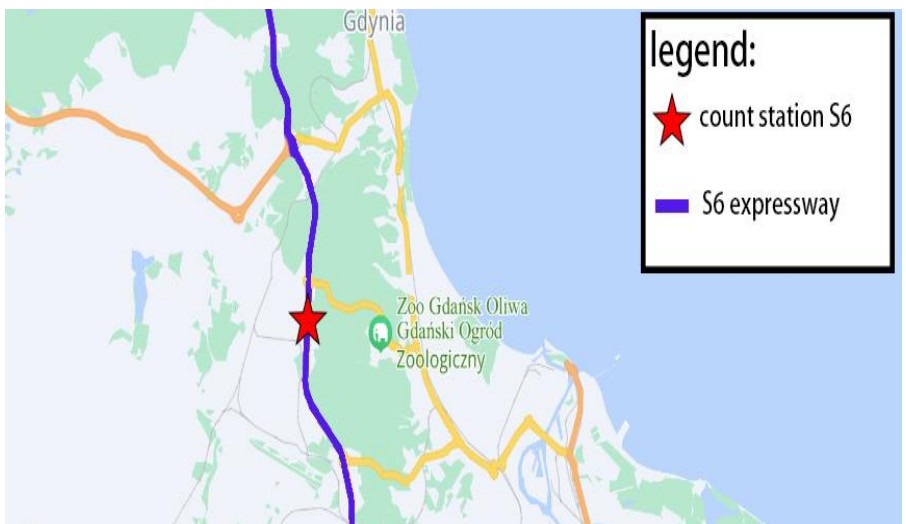

**Figure 4.** Location of the traffic counting station on S6 expressway in Tricity.

The data are aggregated into 5 min intervals. Traffic volume for each time period, rainfall intensity, temperature and whether it was a weekend were used as data for our experiments. However, in the original dataset, the data from 2:50 to 2:55 am on 2 November 2014 was missing, accounting for a mere 0.0165% of the total. And given that it was in the late evening, there were no obvious features. To avoid any impact on subsequent experiments, we opted a relatively simple hot-deck imputation method. For the missing traffic data, data from the same moment in time two days before and after were used to fill in the missing values. Missing temperature and rainfall intensity were filled in with data from before and after. After processing, the data are shown in Table 1. The data used in this study, presented in Table 1, cover the time period from 27 October 2014 to 16 November 2014. "Temperature" represents the average temperature within five minutes; "Rain intensity" ranges from 0 to 100, the larger the value, the more rainfall; and "Working day" is calculated and judged by date: "1" means weekday, "0" indicates the weekend.

**Table 1.** Data sample table.

| Date | Time | Traffic Volume | Temperature | Rain Intensity | Working Day |
|---|---|---|---|---|---|
| 27 October 2014 | 0:00–0:05 | 17 | 11.1 | 0 | 1 |
| 27 October 2014 | 0:05–0:10 | 23 | 11.1 | 0 | 1 |
| 27 October 2014 | 0:10–0:15 | 16 | 11.1 | 0 | 1 |
| 27 October 2014 | 0:15–0:20 | 11 | 11.1 | 0 | 1 |
| 27 October 2014 | 0:20–0:25 | 10 | 11.1 | 0 | 1 |

### 3.2. Correlation Analysis

Temperature, rainfall intensity and whether it is a weekday were chosen as influencing factors for traffic forecasting. In order to explore the impact of the correlation on the prediction accuracy, the Pearson correlation coefficient was adopted to describe the relationship between traffic flow and the variables. It is worth noting that the variable of whether it is a weekday is a categorical variable and the traffic flow data are continuous variables, so the Pearson correlation coefficient cannot describe the relationship between these two variables well. Therefore, we use the Pearson correlation coefficient to describe the relationship between traffic flow and temperature and the relationship between traffic flow and rainfall intensity. The Pearson formula is as follows [31]:

$$r = \frac{\sum\limits_{i=1}^{n} \left( X_i - \overline{X} \right) \left( Y_i - \overline{Y} \right)}{\sqrt{\sum\limits_{i=1}^{n} \left( X_i - \overline{X} \right)^2} \sqrt{\sum\limits_{i=1}^{n} \left( Y_i - \overline{Y} \right)^2}} \tag{18}$$

where $r$ represents the Pearson correlation coefficient, $X_i$ and $Y_i$ are the $i$-th value of variable $X$ and the $i$-th value of variable $Y$, respectively, and $\overline{X}$ and $\overline{Y}$ denote the mean of the variable.

The Pearson correlation coefficient indicates a linear relation between two indicators. It ranges between $-1$ and $+1$ and values closer to $-1$ and $+1$ imply a strong correlation. Also, a positive correlation coefficient implies that an increase in one indicator would result in an increase in another indicator, and vice versa. The relationship between the $r$ value and the correlation strength is shown in Table 2 [31].

**Table 2.** The relationship between the $r$ value and the correlation strength.

| $r$ Value | Correlation Strength |
| :---: | :---: |
| $|r| = 0$ | completely irrelevant |
| $0 < |r| \leq 0.3$ | basically irrelevant |
| $0.3 < |r| \leq 0.5$ | low correlation |
| $0.5 < |r| \leq 0.8$ | highly correlated |
| $|r| = 1$ | completely relevant |

The result of Pearson correlation coefficient analysis is displayed in Figure 5. From the correlation coefficient in Figure 5, it can be seen that the correlation coefficient between traffic flow and temperature is 0.38, and which is relatively low. The correlation coefficient between traffic flow and rainfall intensity is only $-0.21$, implying almost no correlation.

Daily traffic is extracted for autocorrelation. The correlation result is shown in Figure 6. The time range is from 27 October 2014 to 16 November 2014 of which 1 November, 2 November, 8 November, 9 November, 15 November and 16 November are non-working days. The conclusion that there is an extremely strong correlation between working days and working days and the same characteristic between non-working days and non-working days is revealed in Figure 6. Although there is a strong correlation between working days and non-working days, the correlation is reduced compared to the previous two. In addition, the correlation between traffic flow on 10 November and 11 November and traffic flow between non-working days is higher. This observation may be attributed to that fact that 11 November was a national holiday (Independence Day), so it is highly likely many people chose to take an extended holiday. Considering the circumstances, we designated 10 November and 11 November as non-working days for subsequent experiments.

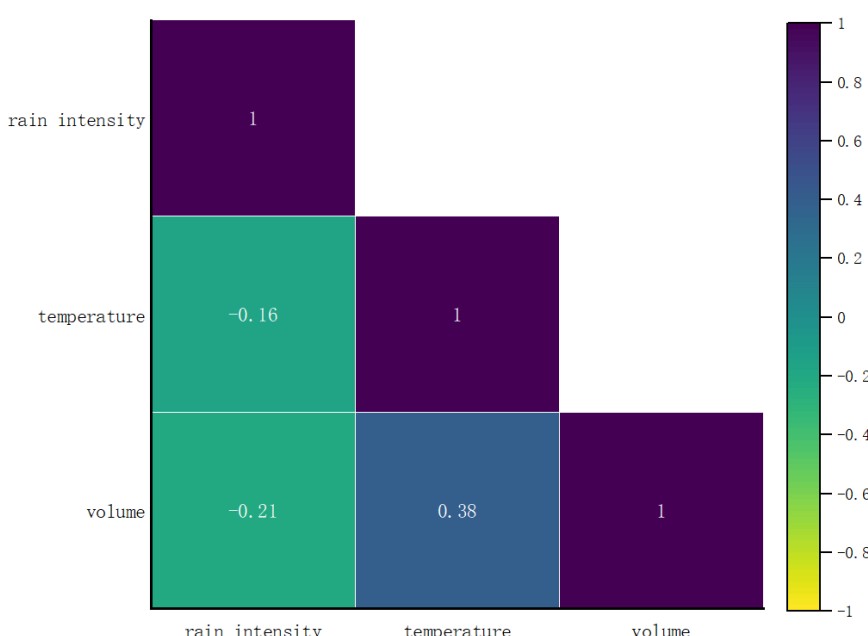

**Figure 5.** The correlation coefficients between traffic flow and variables.

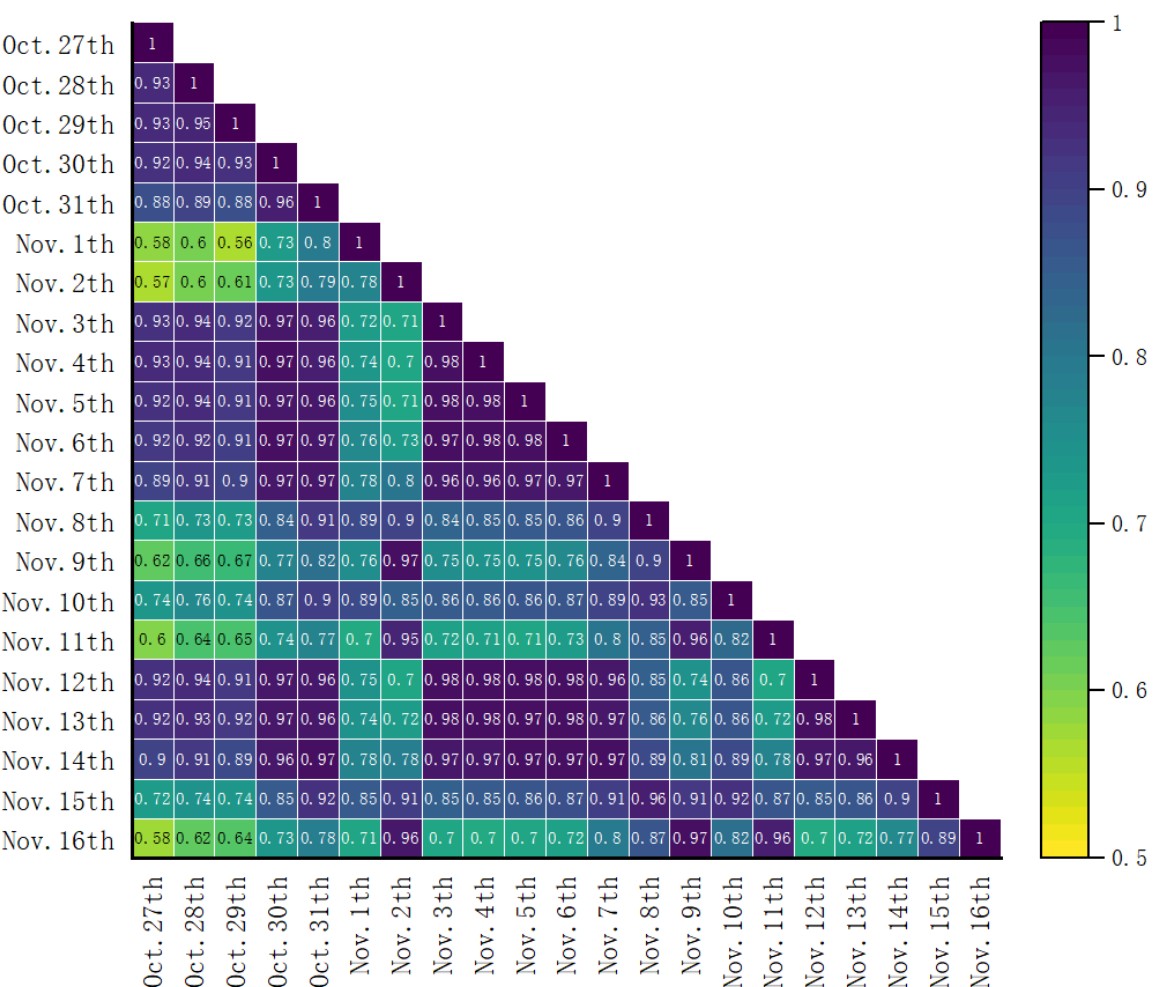

**Figure 6.** The correlation coefficient between daily traffic flow.

## 4. Experiments

In this section, the validity of the proposed models will be explored firstly. Then, we conduct experiments on different missing scenarios considering different factors, and judge the impact of different factors on the prediction based on the experimental results. Finally, predictions are made at different stacking levels to test the effect of stacking levels on the effect of the models.

### 4.1. Missing Data Setting

When traffic flow data are missing, the amount and distribution of missing data can have an impact on the prediction performance. To explore the impact of missing data on prediction, type of missing data and the rate of missing data are set.

In this paper, two types of missing data scenarios are set by us, which are random missing data scenario and non-random missing data. In the case of random missing data, the missing data are random and the missing data do not depend on any variable. As shown in Table 3, this scenario does not have any pattern in the missing rate. The other missing case is the non-random missing case. In this missing case, the missing data depend on other variables to some extent and show some regularity. In this study, we set it as continuous missing in the same time period. As shown in Table 4, this missing case shows as consecutive days of missing data in the same time period.

**Table 3.** Setting of the random missing scenario.

|           | 29 October 2014 | 30 October 2014 | 31 October 2014 | 1 November 2014 |
|-----------|-----------------|-----------------|-----------------|-----------------|
| 0:30–0:35 | 19              | 18              | NA              | 13              |
| 0:35–0:40 | 19              | NA              | 16              | 17              |
| 0:40–0:45 | 7               | 8               | 10              | NA              |
| 0:50–0:55 | 9               | NA              | 9               | 15              |
| 0:55–1:00 | 6               | 9               | 8               | NA              |
| 1:00–1:05 | NA              | 15              | NA              | 11              |
| 1:05–1:10 | 12              | 10              | 10              | 11              |

**Table 4.** Setting of the non-random missing scenario.

|           | 29 October 2014 | 30 October 2014 | 31 October 2014 | 1 November 2014 |
|-----------|-----------------|-----------------|-----------------|-----------------|
| 0:30–0:35 | 19              | 18              | 18              | 13              |
| 0:35–0:40 | 19              | 13              | 16              | 17              |
| 0:40–0:45 | NA              | NA              | NA              | 20              |
| 0:50–0:55 | NA              | NA              | NA              | 15              |
| 0:55–1:00 | NA              | NA              | NA              | 13              |
| 1:00–1:05 | NA              | NA              | NA              | 11              |
| 1:05–1:10 | NA              | NA              | NA              | 11              |

Based on the type of missing data, we set three rates of missing data: 10%, 20% and 30%, respectively.

### 4.2. Parameter Setting

Through iterative testing, the final parameters of the models were determined as shown in Table 5.

In addition to the above parameters, the first 16 days were used as the training set and the last 5 days as the validation set, with mean squared error (MSE) as the loss function, expressed as follows [32]:

$$\text{MSE} = \frac{1}{T} \sum_{t=1}^{T} (\hat{y}_t - y_t)^2 \tag{19}$$

where $\hat{y}_t$ and $y_t$ denote the predicted values and actual values at time $t$, respectively; $T$ is the total number of predicted samples.

**Table 5.** Detailed description of parameters.

| Parameter | Value |
| --- | --- |
| Number of each hidden layer neurons | 24 |
| Training epochs | 50 |
| Activation function of fully connected layer | Tanh |
| Input length | 12 |
| Batch size | 32 |
| Learning rate | 0.001 |
| Optimizer | Adam |

*4.3. Evaluation Metrics*

To evaluate the model performance, three evaluation metrics were used, namely the mean absolute error (MAE), root mean square error (RMSE) and coefficient of determination ($R^2$), which can be defined as follows [33]:

$$\text{MAE} = \frac{1}{T}\sum_{t=1}^{T}|\hat{y}_t - y_t| \tag{20}$$

$$\text{RMSE} = \sqrt{\frac{1}{T}\sum_{t=1}^{T}(\hat{y}_t - y_t)^2} \tag{21}$$

$$R^2 = 1 - \frac{\sum_{t=1}^{T}(\hat{y}_t - y_t)^2}{\sum_{t=1}^{T}\left(\frac{1}{T}\sum_{t=1}^{T}y_t - \hat{y}_t\right)^2} \tag{22}$$

The MAE is the average of the absolute errors, regardless of the positive or negative side of the error, and ranges from 0 to infinity. The MAE is characterized as being relatively insensitive to the point of outliers.

Like the MAE, the RMSE takes on a range of values from 0 to positive infinity; the larger the error, the larger the value of the RMSE. However, RMSE is more affected by outliers.

$R^2$ value closer to 1 means that the prediction is better. If the $R^2$ value is 0, this means that each predicted value of the sample is equal to the mean, exactly the same as the mean model. If the $R^2$ value is less than 0, it means that the constructed model is not as good as the mean model [33]. In the subsequent experimental results presentation, the percentages of $R^2$ will be used.

*4.4. Prediction Results without External Factors*

In this section, external factors are not taken into account in the model. GRU, RNN and LSTM were combined with the estimated model to form the corresponding hybrid models and compared with the proposed models, which illustrate their validity. The prediction module parameters are the same as those in Table 5. All models were implemented with tensorflow and keras framework.

The experimental results in Tables 6 and 7 show the prediction results for the random missing scenario and the non-random missing scenario, respectively. The results show that K-BiLSTM outperforms other models regardless of the missing scenario and missing rate, and K-BiLSTM prediction accuracy is better than the combination of KNN and other models, and P-BiLSTM also shows the same characteristics. The tables revealed that the improvement in prediction accuracy for the K-BiLSTM and P-BiLSTM is not significant at a lower data missing rate. However, at the data missing rate of 30%, the prediction accuracy improvement in both models becomes more prominent. Another conclusion that can be drawn from these two tables is that the prediction error using the same model in non-

random scenarios is slightly lower than that in random missing scenarios with the same missing rate under different missing scenarios. The reason for this phenomenon is that the interpolation module does a better job of completing the data in the case of non-random missing scenario and retains more of the traffic flow characteristics. In the random missing scenario, the prediction effect of the combined model of PMM will be slightly worse than that of the combined model of KNN. This demonstrates that the KNN module can better handle missing data without including external factors. Moreover, the increase in the data missing rate has, to some extent, resulted in a decrease in model prediction accuracy. In the random missing scenario, as the missing data rate increased from 10% to 30%, the K-BiLSTM exhibited an increase in MAE and RMSE by 1.94 and 2.88, respectively, and $R^2$ increased by 1.2%. For the P-BiLSTM, the corresponding increase in MAE and RMSE was 1.75 and 3.8, respectively, and $R^2$ increased by 2.25%. A similar trend was observed in the non-random missing scenario.

**Table 6.** Prediction results under random missing scenario.

| Model | 10% | | | 20% | | | 30% | | |
|---|---|---|---|---|---|---|---|---|---|
| | **MAE** | **RMSE** | **$R^2$** | **MAE** | **RMSE** | **$R^2$** | **MAE** | **RMSE** | **$R^2$** |
| K-RNN | 13.29 | 18.42 | 94.63 | 13.63 | 19.21 | 92.84 | 14.66 | 20.59 | 92.92 |
| P-RNN | 13.32 | 18.23 | 94.45 | 13.83 | 19.16 | 93.85 | 15.77 | 21.87 | 91.84 |
| K-GRU | 13.21 | 18.54 | 94.66 | 13.71 | 19.05 | 93.94 | 15.32 | 21.55 | 91.95 |
| P-GRU | 13.37 | 18.86 | 94.64 | 13.79 | 19.45 | 92.89 | 15.51 | 21.91 | 91.99 |
| K-LSTM | 12.64 | 17.83 | 94.93 | 13.58 | 19.37 | 93.75 | 14.82 | 21.18 | 92.52 |
| P-LSTM | 12.66 | 17.96 | 94.81 | 13.95 | 19.37 | 93.42 | 15.32 | 21.67 | 92.16 |
| **K-BiLSTM** | **12.18** | **17.29** | **95.01** | **13.15** | **18.73** | **94.15** | **14.12** | **20.17** | **93.81** |
| P-BiLSTM | 12.21 | 17.46 | 94.96 | 13.65 | 19.56 | 93.61 | 15.20 | 21.26 | 92.71 |

**Table 7.** Prediction results under non-random missing scenario.

| Model | 10% | | | 20% | | | 30% | | |
|---|---|---|---|---|---|---|---|---|---|
| | **MAE** | **RMSE** | **$R^2$** | **MAE** | **RMSE** | **$R^2$** | **MAE** | **RMSE** | **$R^2$** |
| K-RNN | 12.97 | 17.83 | 94.27 | 13.43 | 19.45 | 93.49 | 14.47 | 21.95 | 91.97 |
| P-RNN | 12.38 | 17.05 | 94.64 | 12.68 | 18.46 | 94.31 | 14.98 | 21.63 | 92.69 |
| K-GRU | 12.53 | 17.67 | 95.02 | 13.98 | 20 | 93.32 | 14.84 | 21.14 | 92.35 |
| P-GRU | 12.34 | 17.55 | 94.97 | 13.74 | 19.52 | 93.27 | 14.03 | 21.02 | 92.62 |
| K-LSTM | 12.17 | 17.19 | 94.92 | 13.26 | 19.39 | 93.73 | 13.90 | 20.67 | 93.04 |
| P-LSTM | 12.31 | 16.91 | 95.22 | 13.19 | 19.24 | 93.82 | 13.95 | 20.62 | 92.91 |
| **K-BiLSTM** | **11.71** | **16.52** | **95.44** | **12.34** | **17.57** | **94.85** | **13.57** | **20.17** | **93.21** |
| P-BiLSTM | 11.94 | 16.8 | 95.29 | 12.51 | 17.89 | 94.66 | 13.69 | 20.40 | 93.06 |

*4.5. Prediction Results Considering External Factors*

In the previous section, predictive performance of K-BiLSTM and P-BiLSTM has been proven. Herein, temperature, rainfall intensity and whether it is a weekday will be added into the model to test the relationship between forecast accuracy and external factors.

Tables 8 and 9 show the prediction results under the random missing scenario and the prediction results under the non-random missing scenario, respectively. The letters at the bottom of the model denote the external factors added, with *z*, *r* and *w* indicating temperature, rainfall intensity and whether it is a weekday, respectively. If there is no letter, it means no external factor is added. Figures 7 and 8 show the prediction errors with different missing rates in different missing scenarios, respectively.

**Table 8.** Prediction results with different alternative combinations of external factors as in put under random missing scenario.

| Model | 10% | | | 20% | | | 30% | | |
|---|---|---|---|---|---|---|---|---|---|
| | MAE | RMSE | $R^2$ | MAE | RMSE | $R^2$ | MAE | RMSE | $R^2$ |
| K-BiLSTM | 12.18 | 17.29 | 95.01 | 13.15 | 18.73 | 94.15 | 14.12 | 20.17 | 93.81 |
| P-BiLSTM | 12.21 | 17.46 | 94.96 | 13.65 | 19.56 | 93.61 | 15.20 | 21.26 | 92.71 |
| K-BiLSTM (*z*) | 13.95 | 18.58 | 94.23 | 14.41 | 20.53 | 92.96 | 15.03 | 21.63 | 92.18 |
| P-BiLSTM (*z*) | 14.73 | 14.72 | 93.34 | 19.87 | 26.43 | 88.34 | 23.90 | 29.46 | 85.51 |
| K-BiLSTM (*r*) | 13.42 | 18.88 | 94.06 | 15.26 | 22.06 | 91.87 | 15.76 | 22.30 | 90.94 |
| P-BiLSTM (*r*) | 14.87 | 20.75 | 92.81 | 17.05 | 24.10 | 90.31 | 21.03 | 28.53 | 86.39 |
| **K-BiLSTM (*w*)** | **11.28** | **15.12** | **96.04** | **11.76** | **17.57** | **95.65** | **11.98** | **17.01** | **95.02** |
| P-BiLSTM (*w*) | 11.85 | 16.44 | 95.49 | 12.21 | 17.06 | 95.14 | 13.06 | 17.57 | 94.85 |
| K-BiLSTM (*z, w*) | 13.13 | 18.36 | 94.02 | 14.23 | 20.28 | 93.15 | 14.99 | 20.77 | 92.89 |
| P-BiLSTM (*z, w*) | 14.55 | 19.86 | 93.41 | 19.03 | 24.91 | 89.64 | 23.25 | 29.17 | 86.72 |
| K-BiLSTM (*z, r*) | 13.33 | 18.59 | 94.23 | 14.90 | 20.75 | 92.81 | 15.35 | 21.66 | 92.17 |
| P-BiLSTM (*z, r*) | 15.4 | 20.96 | 92.68 | 21.1 | 27 | 87.84 | 24.58 | 31.49 | 83.45 |
| K-BiLSTM (*w, r*) | 13.73 | 19.26 | 93.99 | 14.04 | 19.83 | 93.44 | 14.37 | 20.16 | 93.27 |
| P-BiLSTM (*w, r*) | 15.54 | 21.53 | 92.26 | 19.72 | 25.43 | 88.34 | 20.88 | 27.63 | 84.26 |
| K-BiLSTM (*w, r, z*) | 13.25 | 19.25 | 94.82 | 14.62 | 20.45 | 93.04 | 15.03 | 21.34 | 92.47 |
| P-BiLSTM (*w, r, z*) | 15.89 | 21.04 | 92.61 | 19.30 | 25.34 | 89.29 | 26.53 | 33.11 | 82.75 |

**Table 9.** Prediction results with different alternative combinations of external factors as input under non-random missing scenario.

| Model | 10% | | | 20% | | | 30% | | |
|---|---|---|---|---|---|---|---|---|---|
| | MAE | RMSE | $R^2$ | MAE | RMSE | $R^2$ | MAE | RMSE | $R^2$ |
| K-BiLSTM | 11.71 | 16.52 | 95.44 | 12.34 | 17.57 | 94.85 | 13.57 | 20.17 | 93.21 |
| P-BiLSTM | 11.94 | 16.8 | 95.29 | 12.51 | 17.89 | 94.66 | 13.69 | 20.40 | 93.06 |
| K-BiLSTM (*z*) | 13.17 | 18.55 | 94.06 | 13.32 | 19.59 | 93.59 | 14.36 | 21.64 | 92.18 |
| P-BiLSTM (*z*) | 13.87 | 20.72 | 92.84 | 17.07 | 25.65 | 89.94 | 18.57 | 28.64 | 86.31 |
| K-BiLSTM (*r*) | 13.25 | 18.63 | 94.21 | 13.76 | 20.51 | 92.97 | 15.35 | 22.84 | 91.31 |
| P-BiLSTM (*r*) | 13.91 | 19.88 | 93.41 | 15.24 | 24.01 | 90.38 | 18.81 | 29.19 | 85.78 |
| **K-BiLSTM (*w*)** | **11.78** | **16.37** | **95.52** | **12.25** | **16.69** | **95.17** | **12.46** | **17.02** | **95.01** |
| P-BiLSTM (*w*) | 11.84 | 16.73 | 95.24 | 12.35 | 17.26 | 94.99 | 13.26 | 19.30 | 93.78 |

**Table 9.** *Cont.*

| Model | 10% | | | 20% | | | 30% | | |
|---|---|---|---|---|---|---|---|---|---|
| | MAE | RMSE | $R^2$ | MAE | RMSE | $R^2$ | MAE | RMSE | $R^2$ |
| K-BiLSTM ($z, w$) | 13.23 | 18.64 | 94.21 | 14.41 | 19.74 | 92.75 | 15.34 | 21.67 | 92.04 |
| P-BiLSTM ($z, w$) | 13.86 | 20.99 | 92.64 | 18.91 | 28.12 | 86.81 | 22.74 | 33.04 | 81.67 |
| K-BiLSTM ($z, r$) | 13.06 | 18.64 | 94.21 | 14.42 | 20.29 | 93.13 | 15.53 | 22.83 | 91.3 |
| P-BiLSTM ($z, r$) | 15.46 | 22.75 | 91.36 | 19.58 | 28.9 | 86.06 | 22.41 | 33.24 | 81.55 |
| K-BiLSTM ($w, r$) | 13.27 | 19.26 | 93.06 | 14.08 | 20.16 | 92.64 | 14.77 | 22.56 | 91.26 |
| P-BiLSTM ($w, r$) | 14.52 | 21.32 | 92.32 | 17.60 | 24.61 | 89.91 | 21.16 | 30.08 | 84.91 |
| K-BiLSTM ($w, r, z$) | 13.17 | 18.50 | 94.32 | 13.34 | 19.51 | 93.65 | 15.34 | 21.25 | 92.61 |
| P-BiLSTM ($w, r, z$) | 14.34 | 21.58 | 92.22 | 20.39 | 29.09 | 85.88 | 23.42 | 34.61 | 80.01 |

From the experimental results, it can be seen that adding different factors to the model can have a significant impact on the prediction accuracy. The inclusion of temperature and rainfall intensity will reduce the prediction accuracy to some extent, while the inclusion of whether it is a weekday will improve the accuracy of the model. Especially in scenarios where 30% of data are missing at random, the MAE and RMSE of the K-BiLSTM considering whether it was a working day decreased by a maximum of 2.11 and 3.36, respectively, and $R^2$ increased by 1.37%. The MAE and RMSE of the P-BiLSTM decreased by 2.12 and 3.06, respectively, and $R^2$ rose by 2.37%. The reason for this is that the correlation between temperature and traffic flow and rainfall and traffic flow is low, but the inclusion of whether it is a weekday will improve the prediction accuracy due to the strong influence of weekends on traffic flow. Because PMM uses linear regression for imputation, its prediction accuracy decreases more significantly when factors with lower correlations are added. The P-BiLSTM exhibited the most significant decrease in prediction performance in the scenario with 30% randomly missing data, when all factors were considered. The MAE and RMSE increased by 11.33 and 11.85, respectively, while $R^2$ decreased by 9.96%. Similarly, in the scenario with 30% non-randomly missing data, the MAE and RMSE decreased by 9.83 and 14.21, respectively, while $R^2$ increased by 12.17%. Another reason is that the model does not extract enough sample features, resulting in a larger error. In addition, the conclusion that the prediction error increases with the increase in missing rate is further verified in this part of the experiment. Finally, it is clear from Figures 7 and 8 that when the same factors are added at the same missing rate, the prediction accuracy of K-BiLSTM outperforms that of P-BiLSTM regardless of the missing data. This indicates that K-BiLSTM is more suitable for traffic flow prediction with missing data. In Figure 9, the $R^2$ distribution of the predicted results with the inclusion of different variables is depicted. With an increasing missing rate, the inclusion of rainfall intensity and temperature leads to a significant decline in prediction accuracy, particularly pronounced with P-BiLSTM.

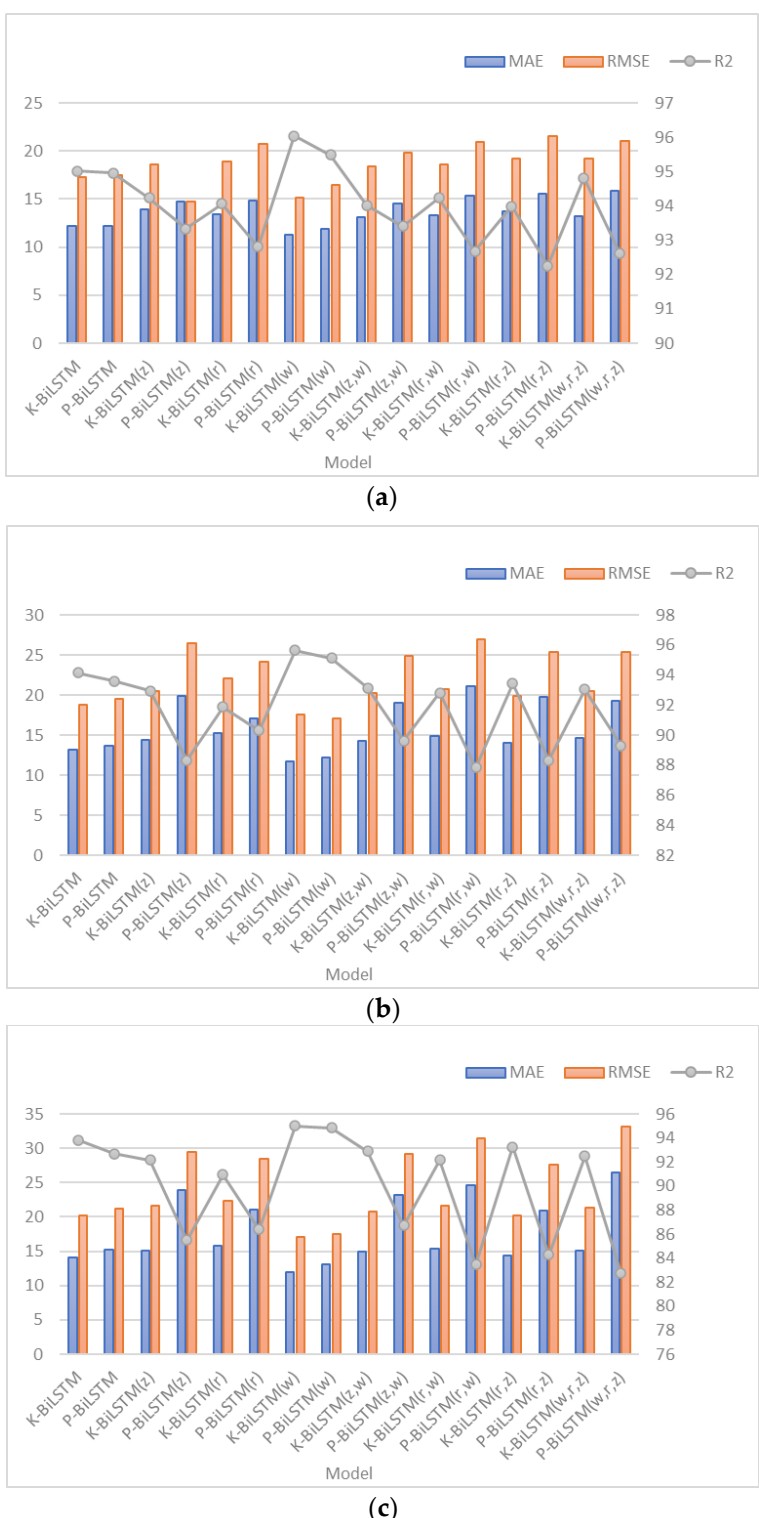

**Figure 7.** Comparison of prediction results considering different combinations of factors under random missing scenario. (**a**) Comparison of prediction results under 10% random missing rate, (**b**) comparison of prediction results under 20% random missing rate, and (**c**) comparison of prediction results under 30% random missing rate.

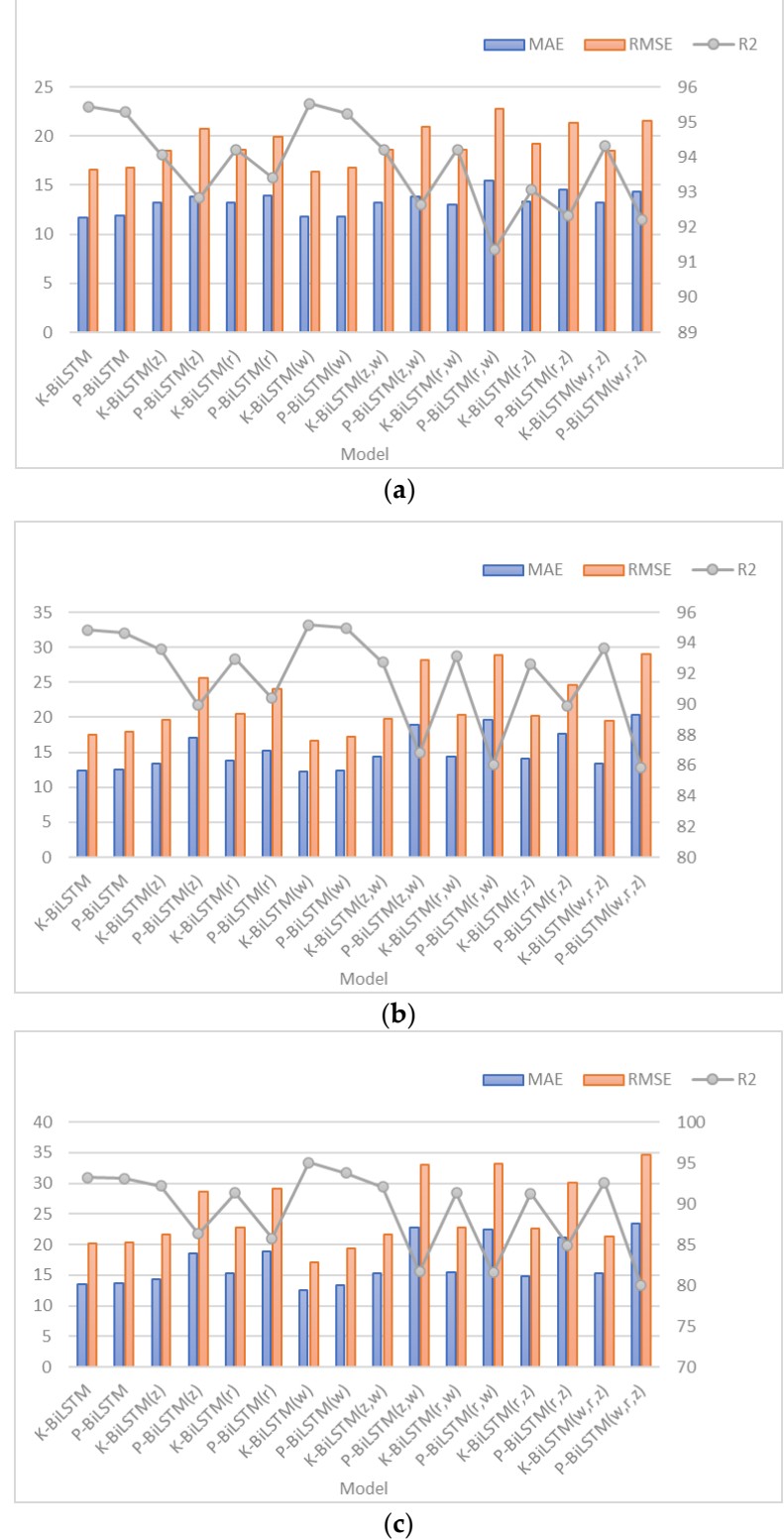

**Figure 8.** Comparison of prediction results considering different combinations of factors under non-random missing scenario. (**a**) Comparison of prediction results under 10% non-random missing rate, (**b**) comparison of prediction results under 20% non-random missing rate, and (**c**) comparison of prediction results under 30% non-random missing rate.

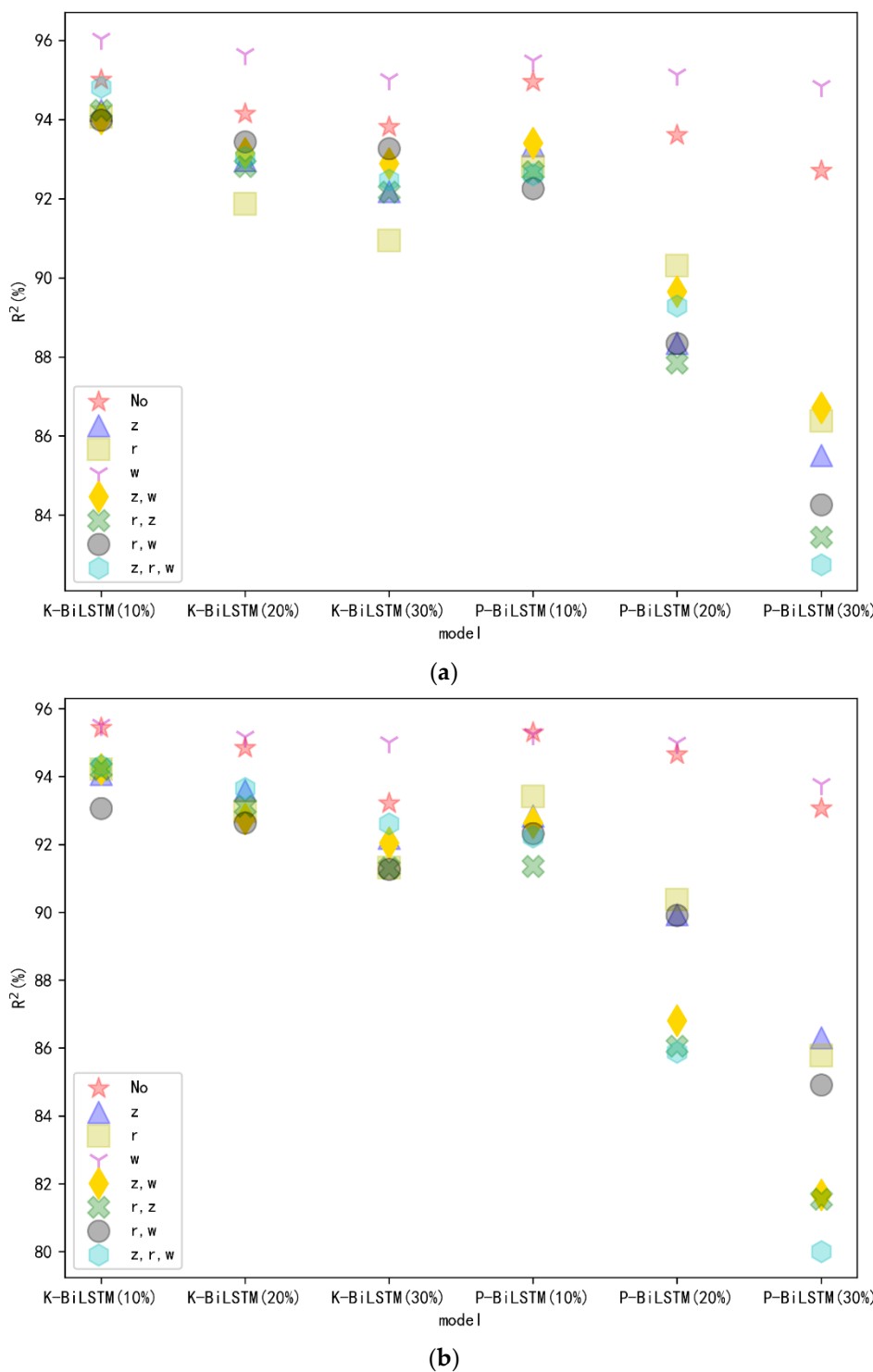

**Figure 9.** Distribution of $R^2$ of predicted results. (**a**) Under random missing scenario, and (**b**) under non-random missing scenario.

## 5. Conclusions

Data loss is inevitable in the process of traffic flow data collection. Therefore, it is necessary to simulate the data loss. Random and non-random data loss scenarios were set up in the experiments. To achieve imputation and prediction, we combined KNN, PMM and RNN, and GRU, LSTM and BiLSTM to achieve data estimation and traffic flow prediction. The K-BiLSTM was experimentally demonstrated to be more accurate than the other models in terms of prediction. In addition, in the experiments where multiple

factors were added to the models, the results showed that performance is only improved by including whether or not it is a working day into the model. Especially in the case of missing 30% of data at non-random, the MAE and RMSE of the K-BiLSTM model, considering whether it was a working day, decreased by a maximum of 2.11 and 3.36, respectively, while the $R^2$ increased by 1.37%. Similarly, the MAE and RMSE of the P-BiLSTM model decreased by 2.12 and 3.06, respectively, and the $R^2$ increased by 2.37%. This relationship was attributed to the correlation between traffic flow data and external factors. The inclusion of factors with low correlation led to an increase in prediction error.

In the future, the following research area will be explored: First, in subsequent studies, the traffic flow prediction can be extended from a single point to the whole urban road network for interpolation prediction. Second, a fusion model that can achieve both imputation and prediction will be studied and proposed. Finally, it is important to improve the application of the model to achieve accurate and efficient prediction under different road conditions.

**Author Contributions:** Conceptualization, W.Z. and R.C.; methodology, W.Z. and K.W.; software, W.Z.; validation, J.Z., K.W. and R.C.; formal analysis, R.C.; investigation, R.C.; resources, J.Z.; data curation, J.Z.; writing—original draft preparation, K.W.; writing—review and editing, W.Z.; visualization, J.Z.; supervision, R.C.; project administration, R.C.; funding acquisition, R.C. All authors have read and agreed to the published version of the manuscript.

**Funding:** This work is supported by the Natural Science Foundation of Zhejiang Province, China (Grant No. LY22G010001) and National "111" Centre on Safety and Intelligent Operation of Sea Bridge (D21013) and the Healthy & Intelligent Kitchen Engineering Research Center of Zhejiang Province and the K.C. Wong Magna Fund in Ningbo University, China.

**Institutional Review Board Statement:** Not applicable.

**Informed Consent Statement:** Not applicable.

**Data Availability Statement:** The data can be obtained from https://mostwiedzy.pl/pl/open-research-data.

**Conflicts of Interest:** The authors declare no conflict of interest.

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
