# Peer review of "Traffic Flow Prediction Based on Hybrid Deep Learning Models Considering Missing Data and Multiple Factors"

_sustainability, doi:10.3390/su151411092_

Round 1

Reviewer 1 Report

Even though the grammar is good, there are a few errata to be corrected, such as 'is to considering' on line 58 and the sentence repetition from lines 34 to 37.

Reviewer 2 Report

The author introduced and validated two models for predicting traffic flow in scenarios with missing data. Subsequently, an experiment demonstrates the relationship between external factors and prediction accuracy by incorporating various factors, and final model stacking experiment confirms the ability of single-layer model to effectively extract data features from the dataset. Traffic flow prediction is a prominent research area in transportation, and data incompleteness is a common issue in traffic data collection. The author integrated the two parts of the study, which holds practical relevance. Therefore, I think this article can be published in this journal if the authors carefully revise it according to the following comments:

(1)    Here are some grammatical and spelling errors, please check and correct them.

For example:

in line 202, is the sentence missing the subject?

(2)    In Eq. (16), whether the input at time  includes data from the historical time period.

(3)    Please add how much precision the model has improved in the conclusion part to highlight validity.

(4)    Please explain in detail that the handling approach for incorporating traffic volumes, temperature, rainfall intensity, and whether it is a weekday as inputs to the model.

Reviewer 3 Report

The authors proposed “Traffic flow prediction based on hybrid deep learning models considering missing data and multiple factors”. In particular, they developed two hybrid models for combining multiple factors for 15 traffic flow predictions in the scenario of data loss. The problem raised in this study could be beneficial for improving traffic flow. However, the authors should work out on the following points

1: The abstract needs to improve/refine. It should be very specific and should indicate the highlight of the proposed work. I found so many redundant information. Also, Grammatical mistakes and/or fragmented sentences. I have listed a couple of them below:

a: During data collection; there is a risk of data loss due to the complexity of the external environment and damage to sensors

b: Experiments have been explored with the traffic flow dataset of the expressway S6 23 in Poland for different missing scenarios with different missing rates.

2: The authors claimed that two hybrid models have been proposed. I don’t know how they integrated the two hybrid models. It is pretty difficult to integrate various parameters/modules of one hybrid model into another. I suggest authors to discuss them comprehensively.

3: Why authors had included the Fig. 1,….,.3. These are the general diagrams. I suggest authors to remove them or draw the proposed model diagram by illustrating them in detail. For example, where they used the LSTM and BiLSTM models in this work, clearly explain them in the diagram.

4: In section 4.1, authors claimed that the data are missing, but how do they know that the data are missing? It is based on assumption? How to distinguish between available data (non-missing data) and missing data. The data needs to train and test, which should be included.

a: They used the PMM and KNN to find the data which is similar to missing data. How? Based on your assumptions? Supervised learning models have better prediction accuracy.

5: In section 4.6, authors claimed that in most studies, researchers stack LSTM and BiLSTM to improve the prediction accuracy where is the references? Authors should add references here

6: In conclusion, authors should remove redundant information. They should merge future research paragraphs.

7: English needs to be improved. 

English needs to be improved as I found many grammatical mistakes and/or fragmented sentences. 

Author Response

See in the attachment.

Reviewer 4 Report

Review Report

Traffic flow prediction based on hybrid deep learning models considering missing data and multiple factors

(ID: sustainability-2411236)

1.      Introduction.

It is claimed that most studies of traffic flow forecasting under missing data "the relationship between data interpolation and external factors has been ignored and the influence of multiple factors has not been fully explored." The authors propose two hybrid models combining multiple factors for traffic flow prediction in the case of data loss.

Multiple factors for data forecasting are Temperature, rainfall intensity and weekday; and Predictive Mean Matching (PMM) and K-Nearest Neighbor (KNN) are introduced for missing value estimation. To improve forecasting accuracy, PMM and KNN were combined with Bidirectional Long Short-Term Memory (BiLSTM) network, as P-BiLSTM and K-BiLSTM to forecast traffic flow.

Case of an expressway in Poland was studied, and its traffic flow dataset for different missing scenarios were analyzed. The models' prediction accuracy were found encouraging.

(In what follows, (x): Line x. Black/Blue fonts are those of the authors/reviewer.)

2.      Minor Comments.

(28) P-BiLSTM and K-BiLSTM may not be considered as proper keywords. So is "Multiple factors."

(167) n1 k 1         Please define n1=?

Take the random number g of … distribution with … degrees of freedom.

Please define the underlying distribution of g. (Normal/Uniform/…?)

(183) The output value of his (?) previous unit Please correct.

(202) In the figure, (missing character?)  is fed into the forward and reverse LSTMs to obtain …which are combined to obtain the final prediction (missing character?). Please correct.

(219) … multiple factors: whether it is a weekday w , the rainfall intensity r and the temperature z .

It seems that Figures 4 and 5 use different notations for rainfall and temperature. Please correct.

(280) …and non-working days is revealed in Figure.3 (?). Please correct the figure number.

3.      Other Comments.

 (57) The most common way to improve the accuracy of model predictions is to considering as many factors affecting traffic flow as possible in the model, such as holidays, weather conditions, road conditions, road pricing policies, precipitation, average wind speed, maximum temperature, minimum temperature and weather types, and static and dynamic factors.

The respectful authors' assertions do not seem to be correct as they found that some of them work to the contrary (the inclusion of temperature and rainfall intensity will reduce the prediction accuracy to some extent). In this connection is the following.

(118) The influence of multiple factors was considered to enhance the interpretability of feature selection. Careful selection of these factors need be embedded in the process. Please address this point within the algorithms.

4.      Concluding Comments.

1.      Review of the literature in the form of writing a sequence of (author name-one expressing sentence, mostly by acronyms) is not a satisfactory way of doing it. I urge the respectful authors to present a critical review of the literature, with categorizations in methods and aspects, such as using multiple factors, and missing data, without resort to acronyms that may not mean anything important to most readers.

2.      The paper is an application of selected sequence of existing methods on predicting short-term traffic volume at one location. This is done using existing packages. It seems that there should be sufficient arguments as to why these packages are used, why are the specifications for the design of the models (among other options) in the package selected, etc.

3.      The presentation of the results are also important to better understanding of the findings of the paper by the readers. Many similar tables and figures with one differing feature, and describing percent of change in several parallel performance measures by different variables (and that in different directions) are confusing to the readers. I urge the respectful authors to design proper means of presenting the findings of the paper.

Author Response

See in the attachment.

Round 2

Reviewer 1 Report

The authors have satisfactorily and thoroughly addressed my misgivings regarding Point 1 and corrected the remaining minor issues.

Thanks to the editor for the opportunity to review this paper, and to the authors for their hard work.

Reviewer 3 Report

The authors have addressed all the comments and incorporated them in the revised manuscript. I found some spelling mistakes in the conclusion part of the response letter. Please double-check the spelling in the revised manuscript to ensure that every phrase is written correctly.

Reviewer 4 Report

Review Report

Traffic flow prediction based on hybrid deep learning models considering missing data and multiple factors

(ID: sustainability-2411236- Revised Version)

1.      Introduction.

The paper has been improved by being responsive to the suggestions made, and correcting the previous errors. However, new ambiguities have arisen, as discussed below, that needs the respectful authors' attention.

(In what follows, (x): Line x.)

2.      Minor Comments.

There are some typos that need be corrected. They include the following:

a.      (263) To avoid any impact on subsequent experiments We opted a relatively simple hot-deck imputation method

b.      (301) Daily traffic is extracted for autocorrelation. The correlation result is shown in Figure.7.

c.       (468) One figure did not exist in the original version (Fig. 9), and here it is replaced by another one. Several figures are changed for untold reasons (current Fig's. 7 and 8), may be for changes in data or figure specifications and captions. Please make sure they are correct representations of the information in the tables.

d.      (555) References need be checked for their numbers.

e.       (499) It can be seen that the prediction results of the multi-layer stacking model are not significantly improved, and the difference between the prediction results and those of the single-layer model is not significant, regardless of how many missing scenarios are missing. The prediction performance of the models with different stacking combinations does not differ significantly, and the prediction results are very close. The stacking model did not significantly improve the prediction performance when the model parameters were already well tuned.

In this paragraph, one point is written in several similar statements.

3.      Other Comments.

(427) The reason for this is that the correlation between temperature and traffic flow and rainfall and traffic flow is low, but the inclusion of whether it is a weekday will improve the prediction accuracy due to the strong influence of weekends on traffic flow.

Please explain why it is worth to do all these analyses for 1 to 2% increase in R2, and to find we have to consider workday data only for prediction of workday traffic, and that temperature and rainfall do not affect this traffic much, particularly that we may use inappropriate factors and become worse off. Please note the following statement in this respect:

(443) In Figure 9, the R2 distribution of the predicted results with the inclusion of different variables is depicted. With an increasing missing rate, the inclusion of rainfall intensity and temperature leads to a significant decline in prediction accuracy, particularly pronounced with P-BiLSTM.

(499) it can be seen that the prediction results of the multi-layer stacking model are not significantly improved, and the difference between the prediction results and those of the single-layer model is not significant, regardless of how many missing scenarios are missing.

Why inconclusive experiments should be presented, particularly that such experiments are not reproducible by others in practice (in the sense that which model's results should be given to what other model).

4.      Concluding Comments.

A.     While I do believe that the technique employed by the respectable authors is a logical and sensible way for data imputation and short-term traffic prediction (that is, traffic flow forecasting with missing data), the way the case study is presented defies the purpose of the paper. The factors that are introduced do the reverse job (temperature and rainfall), and leave the problem with this obvious result that for weekday/holiday traffic data imputation, one should employ the same type of data for estimating the missing data. The latter seems to be a natural factor to choose for any missing data analysis.

B.     Please note that it is hoped that papers such as this paper show the way to do the job more effectively, and without resort to similar extensive analysis. If use of a factor defies its purpose, then it seems that the analysis would be misled. A sound procedure should detect and exclude irrelevant factors in one (not multiple) application(s) of the procedure.

C.     It seems to me that "model stacking" is done to collect a set of different positive aspects of them to take effect in the analysis. This may/should depend on the problem, the case under study, and the available models themselves. Then, the respectful authors should explain, why they are stacking the models they picked, and what do they expect to result from this operation. Only then, what happens may lead to a correct conclusion. If an operation happens to be irrelevant to certain analysis, concluding that model stacking do not have positive effect on the analysis may be misleading.

D.     While I do believe that increase in performance measures at higher levels is much harder than that at lower levels, the 1 to 2 percent improvement in, say, R2 may be very impressive. However, this is not what we are concerned in such studies. The honorable authors correctly point out that they are doing this for a better prediction of traffic volumes, and travel time times, to better manage the transportation network performance (c.f., first paragraph of "Introduction", lines 29 to 37). Therefore, the question arises as to how this 1 to 2 percent increase in the prediction accuracy affects the ultimate purpose of the study.

E.     The respectful authors are aware that it is the prediction of a road network flow that is important in traffic management, and not that of a single road (c.f., line 539). Therefore, the extension of the method from a road to a road network is an important aspect in any such network management scheme.

Round 3

Reviewer 4 Report

Review Report

Traffic flow prediction based on hybrid deep learning models considering missing data and multiple factors

(ID: sustainability-2411236- Revised Version 2)

1.     The respectable authors are expected to exercise sufficient care in the correctness of their paper. Recent responses, after several rounds of review, include (in their words): The original sentence has been revised/The sentence has been modified to read …/In response to comments from other reviewers, we have performed a re-experiment/ the number of references has been modified/ We have trimmed similar parts of the original/ …

2.     The reviewer believes that the technique employed by the respectable authors is a logical and sensible way for traffic flow forecasting with missing data:

(a)   However, the way the case study is presented defies the purpose of the paper. A sound procedure should detect and exclude irrelevant factors in one (not multiple) application(s) of the procedure.

(b)   Next, the question still remains as to how the 1 to 2 percent increase in the prediction accuracy (R2) affects the ultimate purpose of the study, the flow forecasting.

(c)    Moreover, multi-layer stacking needs design. One layer should complement the other to be effective. Simple use of layers by other authors do not warrant replicating irrelevant experiments and present non-conclusive results that may become misleading.

(d)   Finally, as there are concerns on the analysis of one road by the proposed method, speaking of the application of the method on a network, without indicating the avenue leading to this end, seems wishful.

3.     The concerns raised in 2 above, have been pointed out to the respectful authors, but have not received convincing justifications. I urge the respectful authors to, at least, raise these points in the paper and present a plausible justification for them.
